# Changes in Fish Taxonomy Affect Freshwater Biogeographical Regionalisations: Insights from Greece

**Theocharis Vavalidis** [1,2], **Stamatis Zogaris** [2,*], **Alcibiades N. Economou** [2], **Athanasios S. Kallimanis** [3] **and Dimitra C. Bobori** [1]

1   Laboratory of Ichthyology, Department of Zoology, School of Biology, Aristotle University of Thessaloniki, 54124 Thessaloniki, Greece
2   Hellenic Centre for Marine Research, Institute of Marine Biological Resources and Inland Waters, Anavissos, 19013 Attiki, Greece
3   Department of Ecology, School of Biology, Aristotle University of Thessaloniki, 54124 Thessaloniki, Greece
*   Correspondence: zogaris@hcmr.gr

**Abstract:** Freshwater fishes are key indicators for delineating biogeographical maps worldwide. However, controversy in regional-scale ichthyogeographic boundaries still persists, especially in areas of high species endemicity, such as in Greece. One problem concerns the taxonomy of the fishes because there have been extensive changes, mainly due to an increased splitting of species in recent years in Europe. Here, we explore why ichthyogeographic boundary disagreements and uncertainties in region-scale biogeographical units persist. We compare cluster analyses of river basin fish fauna in Greece using two taxonomic datasets: the older fish taxonomy (from 1991) and the current taxonomy that now follows the phylogenetic species concept (PSC), which has become widely established in Europe after 2007. Cluster analyses using the older fish taxonomy depicts only two major biogeographical regional divisions, while the current taxonomy defines four major regional divisions in mainland Greece. Interestingly, some older maps from the pre-PSC taxonomy era also similarly show four ichthyogeographic divisions in Greece and we can assume that the older biogeographical work did not solely use numerical taxonomy but followed an expert-guided synthesis; the older regional definitions have persisted quite well despite radical changes in Europe's fish taxonomy. Through the prism of biodiversity conservation planning, we hope this review may help identify ways to help standardize policy-relevant biogeographical mapping.

**Keywords:** freshwater ecoregions; fish; biogeography; cluster analyses; Balkans

## 1. Introduction

Biogeographical maps are of outstanding importance for conservation evaluations, protected area planning, water management and education [1–3]. Despite two centuries of active interest [4], and recent methodological advances in delineating biogeographic regions [5–7] many map boundary lines often incite controversy (e.g., [8–10]). Freshwater fishes are widely utilized as key biogeography indicators for developing such maps [11–13]. Zoogeographers have long recognised the biogeographic importance of the long-term freshwater-specific isolation of fishes within river and lake basins [14,15]. The delineated regional areas, often called ichthyogeographic regions, are defined by a characteristic pool of freshwater fish species, which are the outcome of long evolutionary histories, climatic refugia, local and global extinctions, and natural colonisation processes [16,17]. Since current fish distributions are well known compared to so many other aquatic organisms, basin area fish assemblages are instrumental for defining freshwater faunal break boundaries [8] and so-called freshwater ecoregional

maps [18]. Classical numerical classification methods, such as cluster analyses, have long been used to quantify levels of ichthyofaunal similarity or dissimilarity among basin faunas in order to define homogenous regions, usually based solely on the presence or absence of species [16]. However, a basis for delineating such ichthyoregions depends on our understanding of species taxonomy, and this has seen rapid changes in recent years.

## 1.1. Fish Species Taxonomy Is Changing in Europe

The species unit is considered as the fundamental taxonomic unit for classification analyses and conservation biogeography [19]. In contrast to many other wholly aquatic organisms, the taxonomy of freshwater fishes is considered rather well studied worldwide [20,21]. Despite this, widespread changes in the taxonomy of fishes have taken place due to the dominance of the so-called phylogenetic species concept (PSC) which is akin to the evolutionary species concept [22] and has replaced the biological species concept in ichthyological publications in the last two decades [22,23]. In the PSC, subspecies do not exist and many former subspecies have recently been elevated to species level.

Although Europe has been researching fish taxonomy for the longest time, taxonomic problems were until recently considered notorious in its inland waters [24–26]. Changes in the European fish fauna nomenclature after Kottelat and Freyhof's Handbook of European Freshwater Fishes in 2007 [22] can only be considered as a taxonomic revolution. They list 546 native species in Europe (excluding Anatolia) whereas wide-ranging checklists before this one listed roughly half the species for the same area (e.g., 213 species and subspecies in Blanc et al. in 1971 [27]). Kottelat and Freyhof's recent list was basically approved by the wider ichthyological community and is widely cited despite some apprehension immediately after publication [23,28]. Both M. Kottelat and J. Freyhof have played an important role in conservation evaluations and biogeographic delineations and related studies though the widespread application of their new species list [29,30]. Following the PSC many European fish species have been re-defined (primarily through examining original species descriptions and "splitting" former widespread species, thus elevating many subspecies to species level). This inflation of "new species" has also increased the use of phylogenetic and other taxonomic interpretations in new checklist compilations [31,32]. However, despite frequent use of a suite of new names, little concern has been published on how changes in the fishes taxonomic resolution may impact biogeographical analyses [8,33].

## 1.2. Biogeographical Delineations in Greece: Why Review Boundaries?

Due to its position among three continents and its diverse and fragmented geography, with long mountain chains and archipelagos, Greece has been a focus area for biogeographic research (e.g., [34–37]). Despite the rich endemic fish fauna [31], in contrast to terrestrial animals and plants, fish-based biogeographic work is rather limited in Greece [38] and many freshwater biogeographical and taxonomic questions are unanswered (e.g., [39]). However, distributional data of fishes from Greece have been used in the wider region for delineating biogeographical units for several decades [11,40,41]. These maps seldom completely agree with each other.

The current fish taxonomy has influenced regional biogeographical boundaries in Europe in a major biogeographical freshwater ecoregion map; that is, the Freshwater Ecoregions of the World (FEOW) initiative [13]. Although FEOW is a biogeographical ecoregion map, it is actually based primarily on the distributions and compositions of freshwater fish species (incorporating the current fish taxonomy in Europe) as well as expert guided major ecological and evolutionary patterns [30]. FEOW is now considered a standard in many conservation and resource management reviews (e.g., [33,38]). Although proposals for incremental changes to FEOW have been proposed, based on our knowledge, no one has ever inquired to see specifically how the newly changed fish taxonomy might have influenced regional-level biogeographic delineations such as these.

We hypothesize that different fish species lists would drastically alter relevant biogeographic regionalisations. The aim of this study is to reveal how the recently evolved new taxonomy of European

freshwater fishes may have affected the discrimination of regional scale ichthyogeographic boundaries. To investigate this, we first apply classical cluster analysis of presence/absence fish data on major hydrographic units (river and distinct lake basins) of Greece using the pre-PSC "old fish taxonomy", based on P.S. Economidis (1991) national checklist [42]. In the same datasheet we run the same analyses with the currently valid names. We are not trying to produce a new ichthyogeographic map or use specialized classification analyses here, we are simply interested in detecting the classification differences primarily due to dissimilar datasets that have evolved from current taxonomic changes. Through the prism of biodiversity conservation planning, we hope this review may help identify ways to help standardize policy-relevant biogeographical mapping.

## 2. Materials and Methods

Freshwater fish-based biogeographic delineations were critically reviewed for Greece. We concentrate our review on published freshwater fish-based delineations which undertake inter-regional ichthyogeographic mapping at a fairly fine-scale "regional level" that is of interest to biodiversity conservation [33,43]). We consider the scale and philosophy of freshwater ecoregions as promoted by the FEOW initiative [30] to be more or less equivalent to our ichthyogeographic regionalization. Other broader continent-wide and global scale ichthyogeographic and biogeographical delineations exist—forming broader and fewer geographical regional units [12,44,45] and these are usually of greater value for coarse-scale or screening-level resource management uses. Finally, we do not examine sub-regional scale biogeography here, although useful work has been developed at the subregional and sub-catchment scale [46–48] and at the phylogenetic level [39] in our wider study area.

We reviewed and compiled native fish lists per river or lake basin totaling 108 major hydrographic units (Figure 1, Table 1) throughout Greece's mainland and also for some major islands. These hydrographic units are defined as semi-isolated and isolated river and lake basin areas; first conceived as inventory units by Economou et al. [23]. We chose to include only hydrographic units where we assume there is a complete native fish assemblage data-set available based both on wide-ranging bibliographic review and recent sampling, including repetitive sampling (in different periods or year) by us and our colleagues [49,50]. Moreover, the field surveys (primarily utilizing electrofishing methods) were based on standardized procedures [51] and sampling usually targeted river bioassesment. Therefore in many areas we are able to have first-hand evidence to interpret which species may be native or translocated by humans [32,52].

**Table 1.** Rivers basins and distinct hydrographic units that are presented in Figure 1.

| | | | | |
|---|---|---|---|---|
| 1. Evros | 23. Aliakmon | 45. Voulkaria | 67. Kotychi | 89. Kifissos (Attιki) |
| 2. Loutros | 24. Vegoritis | 46. Lefkada | 68. Pinios (Peloponnese) | 90. Rafina |
| 3. Apokrimno | 25. Kastoria | 47. Astakos | 69. Alfios | 91. Marathon |
| 4. Samothraki | 26. Mavroneri | 48. Evinos | 70. Neda | 92. Assopos (Beotia) |
| 5. Filiouri | 27. Pinios (Thessaly) | 49. Mornos | 71. Peristeras | 93. Yliki |
| 6. Bospos | 28. Prespa | 50. Assopos Pel | 72. Filiatrino | 94. Kifissos Beo |
| 7. Kompsatos | 29. Aoos | 51. Stymphalia | 73. Yannousagas | 95. Larymna |
| 8. Vistonis | 30. Zaravina | 52. Kandila | 74. Minagiotiko | 96. Platanias |
| 9. Kossinthos | 31. Kerkyra | 53. Feneos | 75. Kleisouraiiko | 97. Sperchios |
| 10. Laspias | 32. Messagis | 54. Krios | 76. Velika | 98. Sourporema |
| 11. Nestos | 33. Kalamas | 55. Krathis | 77. Pamissos | 99. Cholorema |
| 12. Nevrokopi | 34. Pamvotis | 56. Tsivlos | 78. Aris | 100. Kireas |
| 13. Marmaras | 35. Paramythia | 57. Vouraikos | 79. Smynous | 101. Manikiotiko |
| 14. Strymon | 36. Acheron | 58. Keronitis | 80. Ardeli | 102. Dimosaris |
| 15. Rihios | 37. Kalodiki | 59. Selinous | 81. Vassilopotamos | 103. Rigia |
| 16. Volvi | 38. Louros | 60. Meganitis | 82. Evrotas | 104. Lesvos |
| 17. Mavrolakas | 39. Ziros | 61. Phoenix | 83. Dafnonas | 105. Samos |
| 18. Anthemountas | 40. Arethoua | 62. Volinaios | 84. Taka | 106. Ikaria |
| 19. Gallikos | 41. Arachthos | 63. Glafkos | 85. Lerni | 107. Rhodos |
| 20. Doiran | 42. Acheloos | 64. Piros | 86. Erassinos Argolida | 108. Kourna |
| 21. Axios | 43. Vouvos | 65. Prokopos | 87. Nea Kios-Nafplion | |
| 22. Loudias | 44. Vlychos | 66. Vergas | 88. Erassinos Vravrona | |

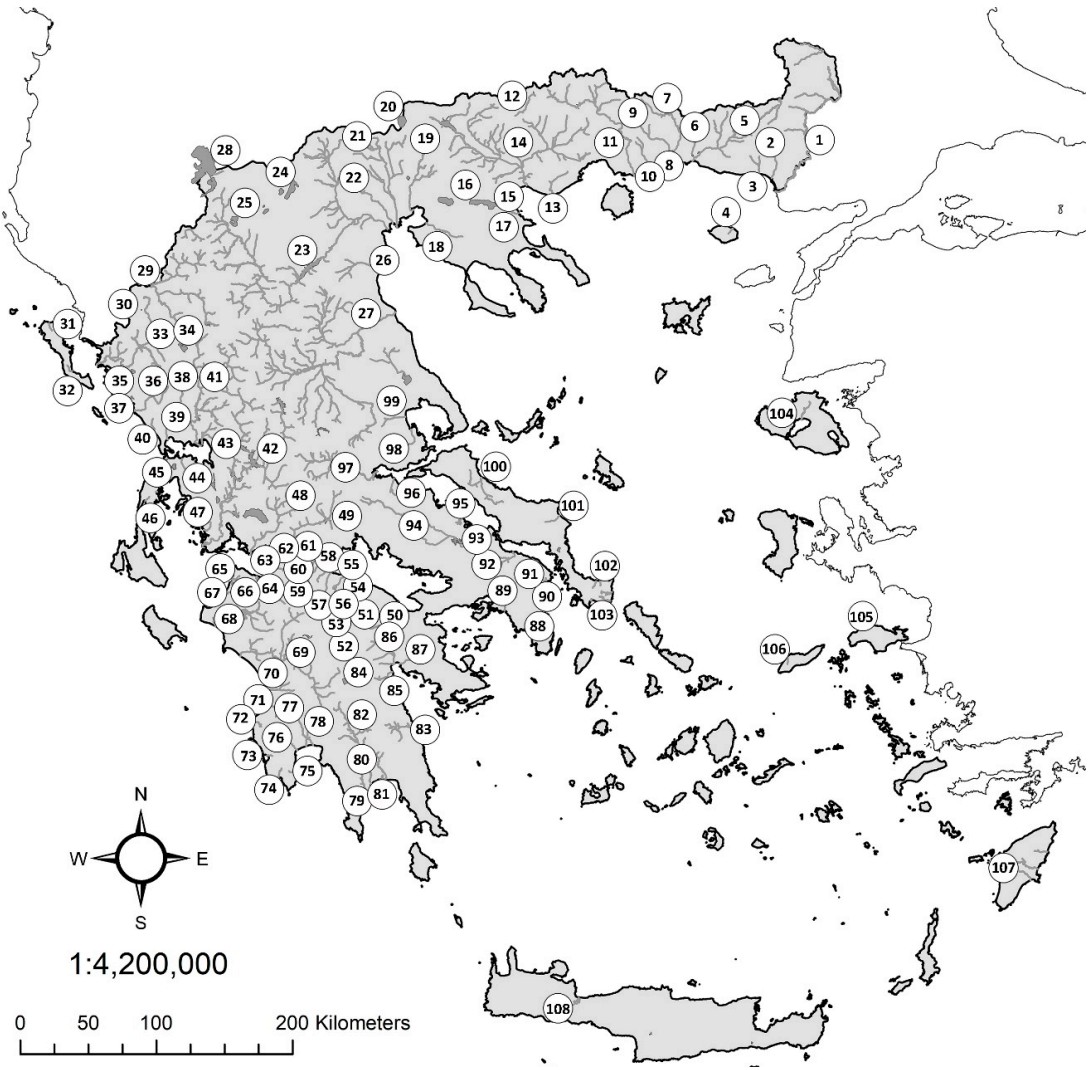

**Figure 1.** River basins and hydrographic units of Greece utilized in the present study.

In our dataset and analysis, alien and translocated species where obviously excluded since they represent anthropogenic effects on species distributions. Moreover euryhaline species (e.g., Mugilidae, *Aphanius fasciatus*, *Atherina* spp.) were also excluded because they may undergo movements in marine waters among basins, which conceals the long term biogeographical isolation effects given by primary and primary-like freshwater species in spatial assemblage patterns [17]. However, a few primary-like species that now lead entire life-cycles in freshwater (i.e., salmonids, sticklebacks (Gasterosteidae) and *Salaria fluviatilis*) have been known to move using marine waters in former times and are included here since they have also been included in older biogeographical analyses [41].

In total 118 species (or equivalent taxonomic units) were included (i.e., 95.2% of Greece's native inland waters ichthyofauna—without considering marine euryhaline species). In this current taxonomy matrix all species reported in the most recent annotated checklist and recent additions are included [31,53]. A second matrix replaced by the older names (following Economidis 1991) was created for comparison (species not listed in 1991 were obviously excluded; i.e., 14 species). The 1991 list was carefully reviewed following the older 1973 catalogue by the same author [54]; one species was included in the old taxonomy though missing in the 1991 checklist (i.e., probably a simple omission error: *Leuciscus* in Prespa).

To apply the cluster analyses, a resemblance matrix was constructed using the Jaccard similarity index, because it has the advantage in presence/absence data not to count for joint absences [55]. Cluster

analyses were determined using group average linkage. In interpreting the resulting dendrograms, initially we used the simplest possible procedure to define major basin-based regional groups: major clusters are simply "eyeballed" at the lowest dendrogram cut-offs for what appear to be discrete and substantial clusters. In order to test the significance of the regional clusters, produced by the resemblance matrix, we used the analysis of similarities test (ANOSIM) [56]. Specifically, we have tested the hypothesis of no differences between clusters at statistically significance level of 0.1%, under 999 max permutations. Moreover, the similarity percentages SIMPER analysis [57], was conducted in order to estimate the average similarity within, and the average dissimilarity between the identified clusters. This also allows a screening of the species' contribution to the cluster composition. All the above analyses were carried out using the Primer 6 statistical package [55]. Finally, our critique concentrated only on major biogeographical boundaries in mainland Greece since the smaller coastal basins and insular basins host idiosyncratic species-poor freshwater fish assemblages [8].

## 3. Results

### 3.1. How Much Has the Taxonomy Changed?

A very high number of fish species have undergone taxonomic changes in Greece since the last published checklist of 1991 (Figure 2). In total, 62 scientific names have changed, 39 have remained the same and 17 were not listed in the previous taxonomy. These name changes are primarily a result of species splitting as promoted by the phylogenetic species concept (for example: *Alburnus alburnus* split to *Alburnus* sp. *Volvi*, *Alburnus thessalicus*, *Alburnus macedonicus*).

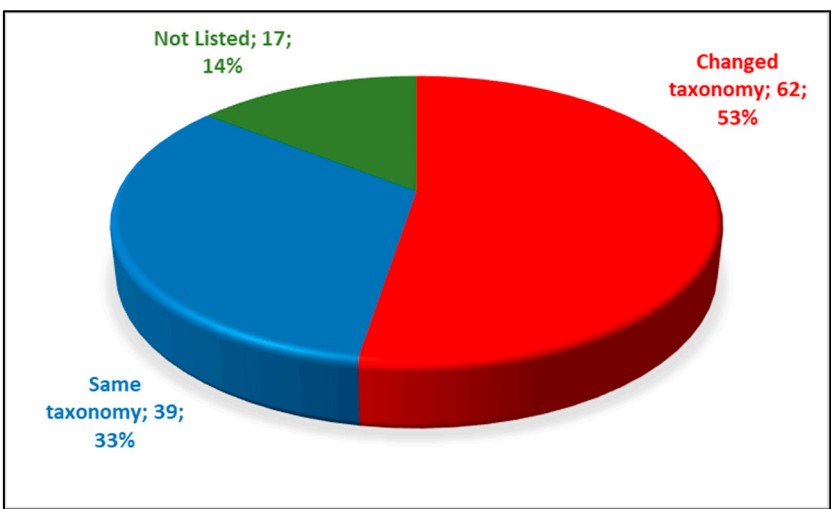

**Figure 2.** Freshwater fish name changes between 1991 to 2019 in Greece (this study's dataset of freshwater species in 108 hydrographic basins).

### 3.2. Different Dataset, Different Ichthyoregions

The cluster analyses show that taxonomic changes would significantly affect freshwater biogeographical boundaries as hypothesized (Figure 3). The pre-2007 taxonomy resulted in two major cluster groups, dividing Mainland Greece into only two biogeographical regions (Figure 3a). There are high-level cut-off subdivisions within these, such as the northwestern basins of Aoos (28) and Prespa (29) which are grouped together. However, within the first major cluster, arbitrarily named here "Northern Aegean", the river basins of Thrace and Macedonia—Thessaly were not clearly separated. The second major cluster contained river basins of the "Greater Ionian", apart from most of the several very small and species-depauperate river basins, e.g., as those on the Aegean island of Euboea (sites 100–103, Figure 1). In contrast, when the current fish taxonomy is used, cluster analysis (Figure 3b) shows that faunal break boundaries are very similar to the current freshwater ecoregion

map (i.e., [38], see discussion). Discrete and biogeographically plausible clusters distinguished four major fish-based ecoregions in Mainland Greece: Western Aegean, Ionian, while there is a separation between Macedonia–Thessaly and Thrace in the third and the fourth branch respectively.

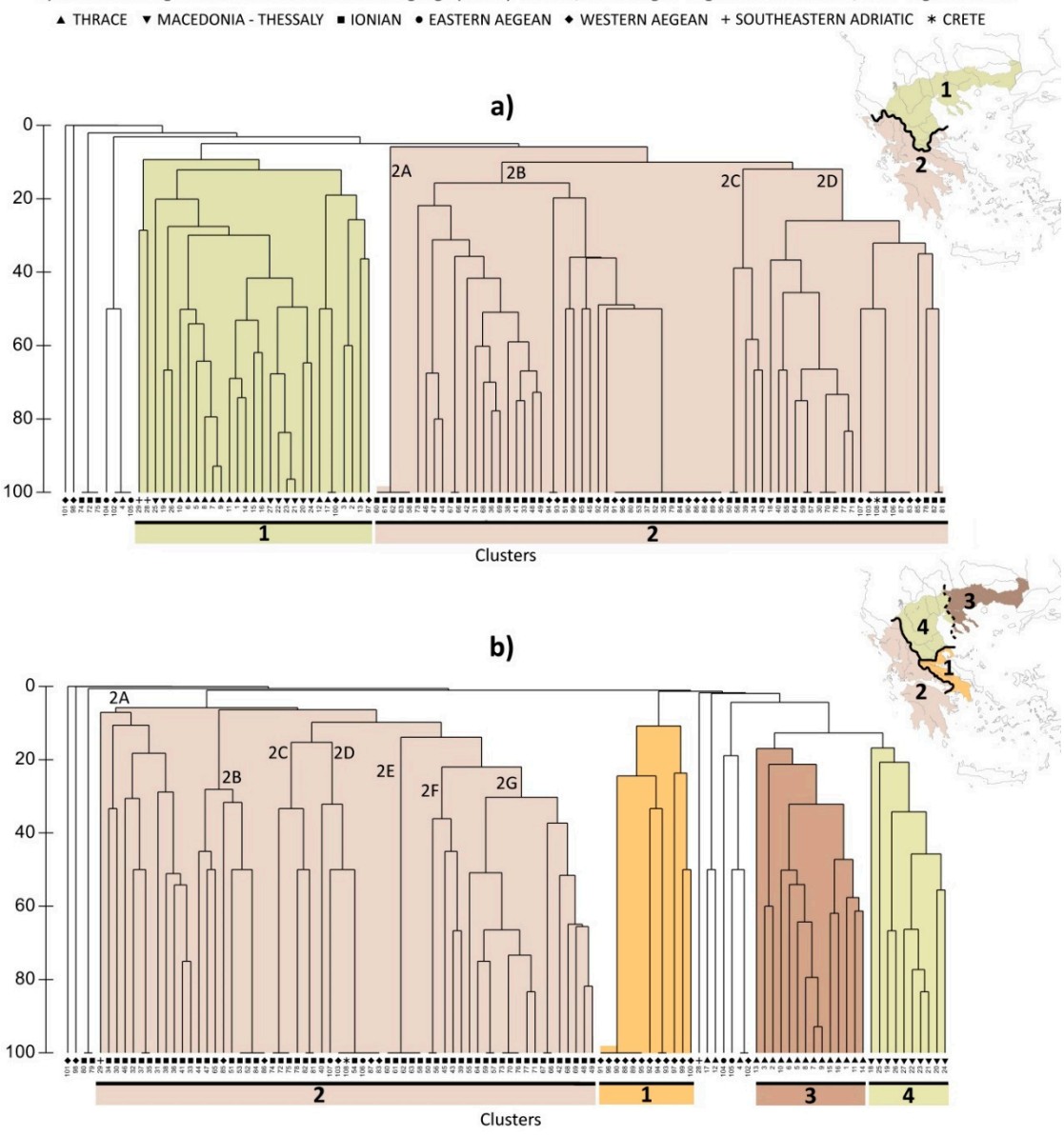

**Figure 3.** (**a**) Cluster analysis according to Economidis (1991) taxonomy. Colored columns show the major regional clusters (1: "Northern Aegean", 2: "Greater Ionian"). The colored codes representing approximately defined freshwater ichthyoregions and numbered sites are mapped (Figure 1). In the coarse inset map, approximate ichthyofaunal break-line boundaries are shown as wide black lines as interpreted from this dendrogram's major regional cluster boundaries. (**b**) Cluster analysis results according to the current taxonomy. Dark columns show the major regional clusters (1: Western Aegean, 2: Ionian, 3: Thrace, 4: Macedonia-Thessaly). In the inset map, approximate ichthyofaunal break-boundaries are shown as wide black lines as interpreted from this dendrogram's major regional cluster boundaries; one boundary is marked in a thatched line denoting comparatively lower significance based solely on this cluster analysis.

According to the ANOSIM test, the probability of observed value in the old (R = 0.40) and current taxonomy (R = 0.44) was less than 0.1% indicating significant differences in both cases for the depicted

regional clusters. ANOSIM results between ichthyoregions are shown in Table 2, only for the current taxonomy, since the old taxonomy cluster revealed no more than two major clusters. In all cases, the significance level was $p < 0.1\%$ indicating significant differences between the observed clusters.

**Table 2.** R values and % significance level of analysis of similarities (ANOSIM) test for the current taxonomy clusters. (Ichthyoregions, as on map in Figure 3b, 1: Western Aegean, 2: Ionian, 3: Thrace, 4: Macedonia-Thessaly).

| Ichthyoregions | 3 | 4 | 2 | 1 |
|---|---|---|---|---|
| 3 | | | | |
| 4 | 0.64 (0.1%) | | | |
| 2 | 0.41 (0.1%) | 0.40 (0.1%) | | |
| 1 | 0.81 (0.1%) | 0.75 (0.1%) | 0.50 (0.1%) | |

We also tested the significance in the differences between subclusters, inside Greece's largest major cluster, the Ionian region, in the current taxonomy (clades 2A, 2B, 2C, 2D, 2F and 2E in Figure 3b). Although we do not investigate sub-regional entities here, we wanted to see if additional "regions" occurred within the Ionian which were not initially evident through eye-balling the cluster (Table 3). The Ionian is characterized by distinctive sub-regions; for example, 2G and 2A, which are statistically significant different at 0.1% with all subclusters of the region. The species that had a high contribution in the differentiation in the 2G subcluster were mainly *Squalius peloponensis* and *Barbus peloponnesius*, both of which are widespread in most of southern part of the Ionian ichthyoregion. For the 2A subcluster *Pelasgus thesproticus* and *Telestes peurobipunctatus* were the most common species of this potential subregion (i.e., the northern part of the Ionian ichthyoregion).

For the old taxonomy, the 2A and 2C subclusters were separated due to low species richness. 2A contained mainly small river basins with *Barbus peloponnesius* and 2C with *Leusiscus cephalus* and *Barbus albanicus*. The 2D subcluster is differentiated from the rest of the Ionian subclusters mainly due to the occurrence of *Salaria fluviatilis* (i.e., presenting a contribution in the similarity of the subcluster of 74.4%).

**Table 3.** R values and % significance level of ANOSIM test for current taxonomy cluster.

| Ionian Subclusters | 2G | 2A | 2D | 2C | 2F | 2B | 2E |
|---|---|---|---|---|---|---|---|
| 2G | | | | | | | |
| 2A | 0.72 (0.1%) | | | | | | |
| 2D | 0.81 (0.1%) | 0.73 (0.1%) | | | | | |
| 2C | 0.82 (0.1%) | 0.83 (0.1%) | 0.82 (0.2%) | | | | |
| 2F | 0.72 (0.1%) | 0.60 (0.1%) | 1.00 (0.3%) | 1.00 (0.2%) | | | |
| 2B | 0.87 (0.1%) | 0.75 (0.1%) | 0.93 (0.1%) | 0.98 (0.1%) | 0.971 (0.1%) | | |
| 2E | 0.81 (0.1%) | 0.68 (0.1%) | 1.00 (0.1% | 1.00 (0.2%) | 1.00 (0.8%) | 1.00 (0.1%) | |

In the associated SIMPER analysis (Table 4) low similarity values were detected in the "Greater Ionian" and Ionian ichthyoregions, in both cases, while the highest similarity value was estimated in Macedonia-Thessaly in the current taxonomy. Beyond these comparisons, high dissimilarity values were assessed between ichthyoregions in both the old and current taxonomy. The lowest value among regions was detected between the current taxonomy's Macedonia-Thessaly and Thrace ichthyoregions (78.78% average dissimilarity) and the highest dissimilarity among regions was found between Ionian and Western Aegean (99.37%).

**Table 4.** The depicted ichthyoregion's similarity (in parentheses) and species percentage contribution, for the old and current taxonomy, according to SIMPER analysis. For the old taxonomy, two arbitrarily named regions are defined. Current taxonomy regional names follow Zogaris and Economou 2017.

| Old Taxonomy | Contrib% | Current Taxonomy | Contrib% |
|---|---|---|---|
| **1. "Northern Aegean" (31.70 %)** | | **1. Western Aegean (45.96%)** | |
| *Leusiscus cephalus* | 17.19 | *Pelasgus marathonicus* | 90.06 |
| *Barbus cyclolepis* | 12.00 | **2. Ionian (20.52%)** | |
| *Rhodeus sericeus* | 9.40 | *Salaria fluviatilis* | 31.71 |
| *Gobio gobio* | 6.22 | *Barbus peloponnesius* | 17.60 |
| *Cyprinus carpio* | 5,76 | *Squalius peloponensis* | 14.49 |
| **2. "Greater Ionian" (27.88%)** | | *Pelasgus stymphalicus* | 13.51 |
| *Pseudophoxinus stymphalicus* | 43.90 | *Tropidophoxinellus spartiaticus* | 5.29 |
| *Salaria fluviatilis* | 19.57 | **3. Macedonia–Thessaly (50.65%)** | |
| *Leusiscus cephalus* | 14.87 | *Squalius vardarensis* | 15.01 |
| *Barbus peloponnesius* | 10.82 | *Barbus balcanicus* | 8.87 |
| *Phoxinellus pleurobipunctatus* | 3.07 | *Cobitis vardarensis* | 7.92 |
| | | *Rhodeus meridionalis* | 7.01 |
| | | *Cyprinus carpio* | 5.47 |
| | | **4. Thrace (48.82%)** | |
| | | *Squalius orpheus* | 16.25 |
| | | *Cobitis strumicae* | 13.09 |
| | | *Rhodeus amarus* | 10.30 |
| | | *Gobio bulgaricus* | 6.89 |
| | | *Cyprinus carpio* | 6.49 |

## 4. Discussion

### 4.1. Different Interpretations, Different Regionalisations

Any kind of numerical analysis of basin fish assemblages obviously depends on the data-set; so ichthyoregions should continue to change if such classifications are based solely on fish species data-sets. However, in the last 30 years, the major fish-based biogeographical maps have not changed drastically after the widespread taxonomic changes in Greece. This is evident when we compare four major fish-based regionalisations (Figure 4): older maps before the taxonomic changes [41,58] and current maps [38,43]. The older maps show four regional entities, strikingly similar to current delineations. However, important discrepancies are evident: (a) there is no boundary showing the Ionian being separate from the Southeastern Adriatic region in the older maps (although at the time, this distinction had been published by Bianco [59]); the two recent maps show a definite "Southeastern Adriatic region"; (b) specifically on the northern Aegean area the two recent map delineations differ; and, (c) within the Southeastern Adriatic region Oikonomou et al. [43] create a new region at the Lake Prespa basin that has never before been defined at a "biogeographical region" level (more on this below).

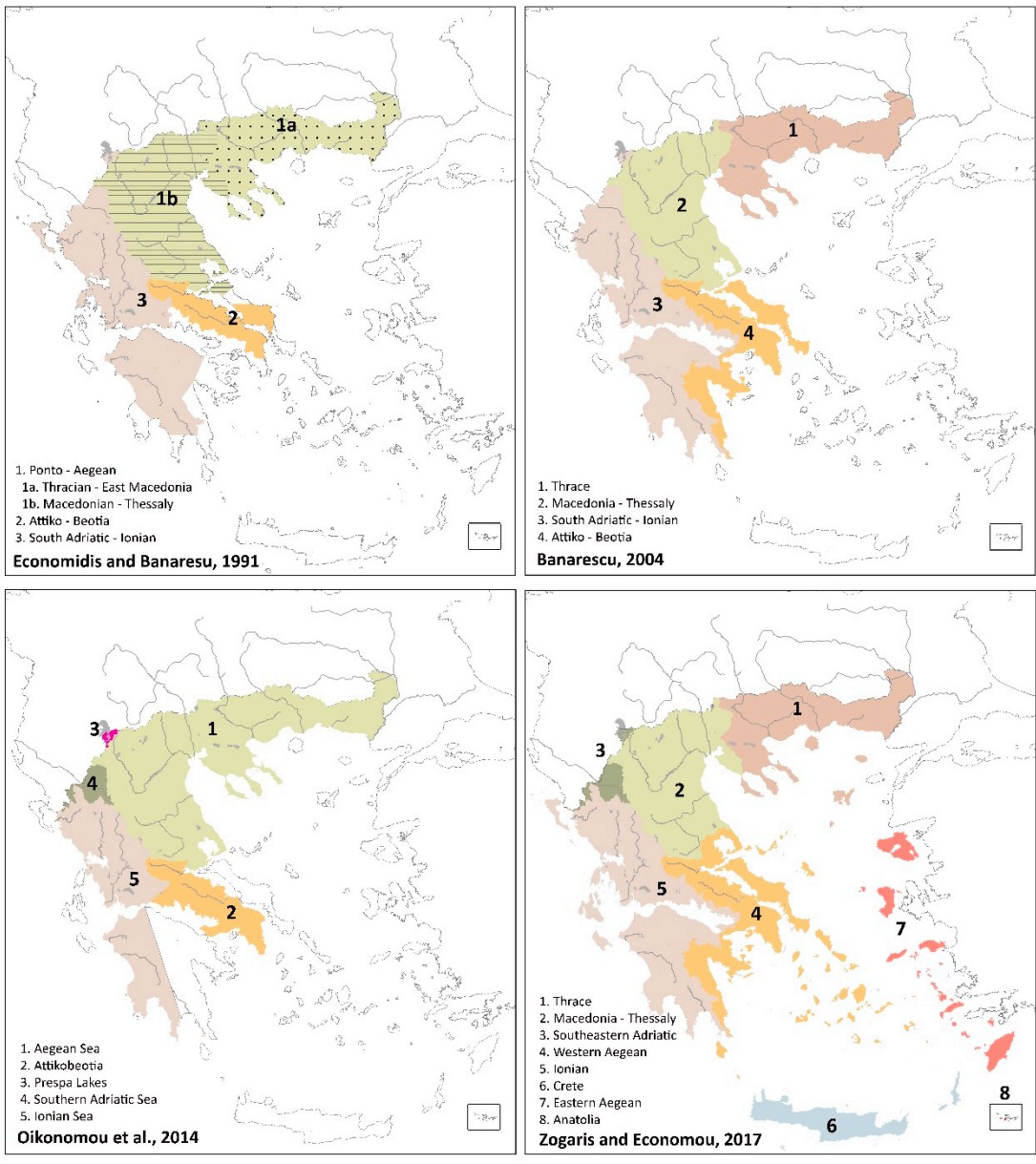

**Figure 4.** Fish-based biogeographic regionalisations from key publications. In the top two maps the old taxonomy was utilized while in the bottomed two the current taxonomy is used.

From a review of the two older maps, we know that in the past, statistical and classification analyses were not dominant as tools to discriminate and interpret biogeographical regional entities; that is, expert-judgment was used through the help of indicator species, as emphasized in Economidis and Bănărescu [41]. It is our opinion that that the four maps depicted in Figure 4 are similar for the following reasons: (a) major delineations are based on long-term geological-geographical barriers to freshwater species dispersal, such as long-standing watershed lines (e.g., the Pindos cordillera) that are well known by biogeographers and dominate in all four maps; (b) indicator species distributions, especially endemic species clusters help define long-standing dispersal boundaries and these were used instead of the full list of native fishes (i.e., the full list has changed, but the ancient endemics form distinct species pools); (c) use of both numerical taxonomy of basin species-lists is often combined with expert-guided interpretations (i.e., incremental adjustments are made but the basic interregional boundary patterns are retained). The retention of the basic ichthyogeographic boundaries in Greece is also testament of a highly distinctive historical biogeography of the wider area (see [60]). However,

despite this, and despite a long period of data gathering and map-making, there are still serious discrepancies among even the two most recent maps (and this includes divergence depicted using the current taxonomy (i.e., in [8,13,43]).

Significant divergence in biogeographic boundary proposals still persist as these are usually based solely on species presence/absence datasets. For example, the Aoos River and Lake Prespa (long known to belong to the Southeastern Adriatic) are not clustered together in our analysis; somewhat surprisingly the Aoos is peripherally pulled towards the major Ionian cluster (Figure 3b). One problem here is that only two river basins from the Southeastern Adriatic are represented in Greek territory and both have high distinctiveness in their assemblages (i.e., many endemics). Beyond this area, a difference between our results and those mapped in Zogaris and Economou [38] concerns the eastern part of Peloponnese (river basins 83 to 87); these clustered with the rest of the Ionian (however, these are very small species-poor river basins that cannot easily be compared with all others). Likewise, several small river basins do not couple with major clusters; this could be a result of depauperate isolated basins on islands and peninsulas, very small river basins with naturally impoverished faunas and/or possible anthropogenic extinction of species in some basins. Finally, there are still specific boundary problems in the northern Aegean area, some analysts have promoted one ecoregion unit there [8,43] whereas others have promoted two; based on the use of similar fish-based analyses [13,32,38]. Based on recent reviews and this current work, we are certain the northern Aegean area comprises two separate and distinct ichthyogeographic regions [13,32,38]. Our interpretation is also supported by the two older biogeographic studies as depicted in Figure 4.

### 4.2. Biogeographical Numerical Taxonomy: The Devil Is in the Details

During the last three decades, most freshwater zoogeographers concentrate solely on quantitative measures, i.e., using a similarity index or multivariate similarity measure to detect areas of marked change for charting break-lines among examined basin faunas [16]. New analytical methods, involving cluster and network analyses among others have helped to quantify levels of faunal similarity among basins, and these are still evolving rapidly [61,62]. However, strictly applying numerical taxonomic methods on species units for such delineations is not a panacea [8] and although there are many methods to produce species-inferred regional-scale maps, opinions differ on best acceptable approaches [63]. A potential pitfall is to focus only on presence/absence species dataset ignoring other variables such as the fauna's phylogeography, species abundances and the historical geological and climatic processes that may have shaped the basin assemblages. Furthermore, the set of species used in the analyses may hide unexpected biases. For example, species of marine-origin that may have actually colonized inland water basins through the sea in different geological periods (unlike no other primary freshwater fishes). In our data set, the widespread *Salaria fluviatilis* (although now confined to inland waters) provides such uncertainty, potentially creating noise in the primary/primary-like freshwater fish assemblages.

The other problem concerns the use of the "species" level unit without reference to phylogenetic relationships (see [64]). When species are treated as if they are equally distinct (as in traditional resemblance measures), then splitting species into finer units will always significantly increase geographical dissimilarity. Most obviously, since species are related to each other in varying degrees, this pattern of relatedness is of substantial importance in interpreting biogeographical patterns. In this way, a phylogenetically-weighted resemblance measure not only reveals deeper evolutionary patterns but is less sensitive to changes in taxonomy (i.e., recently split taxa will still be shown to be closely related) (e.g., [65]). Whatever the numerical and biogeographical analyses used in exploring inter-basin faunal relationships, uncertainties will be better interpreted through combining quantitative analyses, phylogenetic approaches and other multi-disciplinary approaches [18,66]. Therefore, a mixed-method approach to fish-based biogeographical regionalization (integrating quantitative and qualitative information) should help guide final boundary decisions [38].

### 4.3. Beware of Anthropogenic Translocations and Faunal Homogenization

Human induced translocations and local extirpations may be difficult or even impossible to ascertain. Numerical classification analyses using fish are sometimes vulnerable to uncertainties in confirming natural native species distributions. This is a widespread problem with terrestrial animals in the Mediterranean, especially its islands [67]. Unlike terrestrial animals on islands, the instances of human-transported fishes are certainly much less frequent in Greece's river basins [52]. However, we have identified a few questionable distributional records in our data set and are still researching some of them. Examples exist where even a single species may have significant impact to biogeographical classification analyses: the peculiar record and unsolved provenance of *Pelasgus thesproticus* in the Aoos River is one such case. This species has only recently been found in the Aoos (in the Albanian section, and in certain locations only [68]) and including it in cluster analyses has a strong influence in increasing the distance from similar Southeastern Adriatic basins such as Lake Prespa. Other such cases include the influence of other easily translocated fishes such as trout, carp, and small gobies (see [69]). Furthermore, some species that have become extinct due to anthropogenic pressures and changes may easily go unrecorded (e.g., [70]). We have documented a few such extirpations, but much historical work to review this issue remains undone [23]. Moreover, guidance for regional boundaries in the species-depauperate islands is not considered here due to their remarkably poor freshwater fish faunas and their unique socio-ecological and cultural conditions that may have increased anthropogenic extinctions [8]. Islands are generally prone to very high extinction rates in many terrestrial and inland water animal species (e.g., [70,71]) thus hosting unique and idiosyncratic assemblages. In the same vein, some very small river basins on the mainland may be structured as "virtual islands"; these basins are often the outliers in such numerical classification analyses.

Genetic screening and careful historical evidence-gathering is required to identify human-introduced species and the relevance of any "missing species". Some historical species records are lost or forgotten: efforts must be made to include the presence of extirpated species where we know for certain they once existed [31]. Mylonas [72] was one of the first to mention the "influence of man" in terrestrial zoogeography of mollusks in the Aegean islands and this has been corroborated in later researches with reptiles [73,74] and mammals [75] in the Aegean. Fishes, could also be transported by humans (especially from lake-to-lake due to lacustrine fishery traditions), so extra care is needed to detect such situations of so-called translocated species from other parts of the country. Furthermore the issue of translocated species (i.e., native species to Greece translocated to non-native territory) is spreading in Greece [52]; and this is a true threat following a biogeographical homogenization trend taking place worldwide [76].

### 4.4. The Issue of Spatial Scale

Biogeographical regions that contain species pools of near-homogenous fish fauna can of course be examined at different spatial scales. Spatially broad or global scales for example, show only 8 fish-based biogeographic regions in Europe (with two in Greece) [12]; in contrast, sub-continental scale analyses show 8 such regions within Greece's territory [13,38]. There is still no standardized terminology specifically limiting the notions of "region" or "subregion" among inland water zoogeographers [17]. Perhaps coarser global approaches (e.g., [12,17]) could best be termed "super-regions"; most freshwater biogeographical ecoregional analyses are usually on narrower regional frameworks [58,66,77–79]. Finally, some of the most rigorous biogeographical regional delineations provide a multi-tiered and multi-scaled hierarchical approach to region delineation, in which associated subdivisions (i.e., subregions) are charted [78,80].

The size limit of an ichthyogeographic region also varies among authors. Conservation practitioners propose that biogeographical regions should be sufficiently large and biologically complex to justify dividing them further into subregions or key conservation subregions [66,81]. As depicted in Figure 4, exceptions to the usual size of ecoregions are located even in the recent fish-based delineations in Greece [43]. Based solely on the numerical analyses, i.e., "high compositional

dissimilarity" of the fish fauna, Oikonomou et al. [43], suggested that the Lake Prespa basin (covering a mere 2520 km$^2$) be delineated as a unique regional entity. Although this area is a hotspot for endemic species, we disagree with the delineation based both on the proposed region's very small size and the proven phylogenetic relations of this area's fish species pool with the wider Southeastern Adriatic ecoregion [64]; previous fish-based biogeographical reviews promote its relation with the Southeastern Adriatic as well (e.g., [38]). Attempts to reach consensus on delineation decision-making in such hot-spot cases, including the minimum size of biogeographic regions, should be further investigated.

*4.5. Regionalisation Prospects*

Biogeographical regional delineations using freshwater fishes helps us to re-focus on historical biogeography baselines: on "where species currently live, have lived in the past, and would live in the future" [82]. Basin-level assemblage data help define regional ichthyogeographic frameworks that underpin conservation schemes; and they are worthy of careful evidence-based review [78]. Fish-based regionalisations are important for developing inland water typologies for water management and monitoring [8], for conservation evaluations [1,78] and for tracking the spread of alien and translocated freshwater species [52,76]. Biogeographic regions define the native regional species pool in a broad sense [83] and provide reference baselines for policy-relevant conservation and restoration planning applications, such as climate-change adaptation, including assisted migration (species re-introduction) for threatened species as well [84]. Since European inland waters are changing quite fast [85], tracking biogeographical baselines is important and this is why we see a recent trend in using biogeographical freshwater ecoregions in many conservation evaluations [86]. However, inconsistency in published map boundaries and an ever-changing regionalisation scheme may negatively affect policy-relevant conservation management and monitoring [8].

Through our comparison at the national scale, we scan for potential repercussions of the recent "fish taxonomic revolution" that has taken place in Europe [22]. It is important for ichthyologists and relevant practitioners to appreciate the nuances of the evolving taxonomy, to continue to contribute openly to fish distributional knowledge [31] and to use the current taxonomy with care in any new regionalisation applications. Moreover, the current fish taxonomy provides many conservation opportunities because it has brought many less-regarded former "subspecies" to the conservation arena [25,32,87]. In our review in Greece, the current taxonomy does corroborate the broadly fish-based biogeographic freshwater ecoregional boundaries as well [30]. In addition, we have shown here that zoogeographers were well aware of these major regional biogeographic boundaries before the current taxonomy was used in species-based numerical classification analyses. Finally, the changing species name issue will continue to be important. Even though Southeastern Europe is relatively well studied for its freshwater fishes, taxonomic uncertainties and species systematics are an active research area for many families of fish in Greece [53,69] and the evolving taxonomy will continue to pose biogeographical boundary questions. We strongly urge the integration of phylogenetic information to promote the use of relatedness among species within more holistic mixed-methods biogeographic analyses [38,39,64,65].

Comparing numerical classifications using fish species distributions among recently-changed nomenclature sets has to our knowledge never been attempted before in regionalisation critiques. Though there can be no perfect regionalisation that satisfies all aspects of biogeographical interest [5], efforts must be made to ameliorate working frameworks, such as the conservation-relevant ichthyoregions and associated freshwater ecoregions. We have shown that the changing taxonomy is not the only force "destabilizing" current biogeographical regionalisations. Important steps towards more rigorous biogeographical analyses and multi-disciplinary approaches should help in successfully delineating regions—in order to support the conservation of natural biogeographical patterns.

**Author Contributions:** Conceptualization, T.V. and S.Z.; Data curation, T.V., A.N.E. and S.Z.; Formal analysis, T.V. and S.Z.; Methodology, T.V.; Supervision, S.Z., A.N.E., A.S.K. and D.C.B.; Validation, T.V., A.N.E. and D.C.B.; Writing—Original draft, T.V. and S.Z.; Writing—review & editing, S.Z. and T.V.

**Funding:** This research received no external funding.

**Acknowledgments:** Field data were collected through several projects at the Institute of Marine Biological Resources and Inland Waters, HCMR (2002–2018). We are especially grateful to Nicholas Koutsikos and Vassilis Tachos who greatly assisted in distributional database development. We are also very thankful to P.S. Economidis and many colleagues who have assisted us in biogeographical research and discussions, particularly and all HCMR ichthyologists who have supported relevant field and lab work. Finally, the reviewers of this paper significantly assisted in the amelioration of clarity and presentation of this research.

**Conflicts of Interest:** The authors declare no conflict of interest.

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
