# Peer review of "Changes in Fish Taxonomy Affect Freshwater Biogeographical Regionalisations: Insights from Greece"

_water, doi:10.3390/w11091743_

Round 1

Reviewer 1 Report

The paper describes how the taxonomic resolution of fish taxa impacts the definition of freshwater biogeographic regions. Results show that finer the taxonomic resolution, finer the resulting biogeographic information. Moreover, you also show that experts in ichthyology have relatively good empirical knowledge about biogeographic regions.

I find the paper well written and figures nicely illustrate the text. Although I feel the main result (finer the taxonomic knowledge, finer the resulting bioregions) is relatively obvious and not particularly earth-shattering, I think you made a thorough and interesting demonstration.

Nevertheless, my main concern deals with the lack of statistical analyses to test if the bioregions identified in figures 3a and 3b are significant. This could be easily tested with ANOSIM, for instance, which is available in the Primer package.

For instance, in figure 3b, similarity among clusters in bioregion 2 is sometimes lower than among your bioregions 3 and 4. Perhaps bioregions 3 and 4 are not significantly different or additional bioregions could be identified in cluster 2 (after having taken into account some outliers)?

Other remarks:

-          P5L143: Simper analysis: A few key explanations about this analysis should be added in the Materials and Methods section.

-          Number of species. A discrepancy in the number of species included in the analysis seems present: Line 134 (112 species) and Line 150 (110 species= 55+41+14). Please can you check or rephrase?

-          Figure 2: the number of species could be added along with “changed taxonomy 49%”

-          Figure 3: Replace ichthyfaunal by ichthyofaunal (twice).

-          In Section 4.1 (L.208-239), the figure number is incorrect. Replace “Figure 3” by “Figure 4” (twice).

-          L. 250-252: “A potential pitfall is to focus only on presence/absence fish dataset ignoring other factors such as species abundances and the historical biogeographic, geological and climatic processes that may have shaped the basin entities at broad regional scales.” I suggest rephrasing the sentence because, from my point of view, species occurrence and abundance are variables, not factors.

-          L. 258-260: “Controversy and uncertainties still persist concerning specific boundaries and freshwater ecoregion entities is based on subtle presence/absence differences and the regional breadth of the regionalisation analyses (i.e. Greece or a much wider area covered).” The sentence is difficult to understand, I suggest rewording.

Author Response

We thank this reviewer for highly constructive commets and have followed his/her suggestions specifically, such as to test if the bioregions identified in figures 3a and 3b are significant using ANOSIM. Although we are not trying to produce a map of regionalization we’ve applied your suggested method in our results since we believe this strengthens the outcome of the results.

Also concerning line   P5L143, a  few key explanations about this analysis are added in the Materials and Methods section.

Also an important correction was made concerning a typographical mistake on the total number of species (i.e. Line 134 (112 species) and Line 150 (110 species= 55+41+14). This was corrected.

Other specific suggestion incorporated include: a)     Figure 2: the number of species could be added along with “changed taxonomy 49%”, b)       Figure 3: Replace ichthyfaunal by ichthyofaunal (twice), c)       In Section 4.1 (L.208-239), the figure number is incorrect. Replace “Figure 3” by “Figure 4” (twice).

Finally, in    L. 250-252 and L. 258- 260 : A rephrasing the sentence has been done.

Reviewer 2 Report

This is a useful study of the effect of taxonomy on biogeographical classifications.

Splitting taxa obviously inflates pairwise dissimilarity of assemblages and this will affect cluster analyses. In general, it should make classifications less stable because the dataset effectively has higher dimensionality. In the extreme case of all pairwise comparisons having maximum dissimilarity, any resulting cluster analysis would be resolved entirely randomly. When species are treated as if they are equally distinct (as in traditional resemblance measures), then splitting species into finer units will always significantly inflate dissimilarity. In reality, species are related to each other to varying degrees and this pattern of relatedness is of substantial importance in interpreting biogeographical patterns. It follows that a phylogenetically-weighted resemblance measure not only reveals deeper evolutionary patterns but is less sensitive to changes in taxonomy (because split taxa will still be closely related). See the following paper for a method of devising phylogenetically-weighted resemblance measures:

Nipperess et al. (2010) Resemblance in phylogenetic diversity among ecological resemblance. Journal of Vegetation Science 21: 809-820.

Note that I'm not proposing that you re-analyse your data with this method as it is likely that phylogenetic data are not available, especially for two separate taxonomies. Nevertheless, I think it is definitely worth including in the discussion the possibility of using phylogenetic resemblance as a means of addressing the problem of splitting taxa.

The choice of major regional clusters seems to have been done arbitrarily. That is, the graph was "eyeballed" for what appeared to be discrete and biogeographically plausible clusters. I'm not necessarily opposed to this approach although I think you should explicitly state that this is what you did. Also, there appears to be no testing of the quality of the classification or the statistical significance of the chosen regional clusters. It is simple enough in PRIMER to calculate the cophenetic correlation, which tells you how closely the cluster dendrogram corresponds to the original resemblance matrix. The statistical significance of the regional clusters could be tested using ANOSIM in PRIMER.

The SIMPER analyses are not described but just mentioned in passing. Please briefly explain in the methods what the SIMPER procedure does and provide a reference if possible.

The quality of writing is generally very good although I did pick up a few errors:

line 59: "recently considered" rather than "considered recently"

line 109: "useful work has been developed"

line 135: this isn't a functional sentence. I'm not sure what meaning was intended. Please review.

line 143: SIMPER should be capitalised because it is an acronym. Check all instances.

line 163: "roughly corresponding to recent delineated regions"

line 187: "One problem"

line 249: "on the best acceptable"

line 259: "entities are based"

line 282: meaning of "quantifies distance" is not clear. I assume you mean that you mean this species has a strong or undue influence on distance?

Author Response

We sincerely thank this reviewer for highly constructive comments and have followed his/her suggestions specifically.  In fact we have incorporated important aspects provided by the reviewer in the text (i.e. with concerns for phylogenetically-weighted resemblance measure, e.g.; see lines: 331-345). We have also included the suggested reference (Nipperess et al. (2010) Resemblance in phylogenetic diversity among ecological resemblance. Journal of Vegetation Science 21: 809-820.).

Importantly we have also chosen to insert the method of cut-offs of the dendrogram, exactly as phrased by this reviewer and we are very appreciative of his comments (i.e. eyeballing: see Methods section Lines: 161-162. Also SIMPER analysis is better expressed in the Methods section as suggested by the previous reviewer:

The following errors cited by this reviewer here corrected:

line 59: "recently considered" rather than "considered recently"

line 109: "useful work has been developed"

line 135: Sentence on “marine movement” of certain non-primary freshwater fishes re-phrased.

line 143: SIMPER capitalized in all instances.

line 163: "roughly corresponding to recent delineated regions"

line 187: "One problem"

line 249: "on the best acceptable"

line 259: "entities are based"

line 282 (now on line 337): Rephrased exactly as suggested.

Reviewer 3 Report

MAJOR COMMENTS

This is an interesting study about fish taxonomy and how its changes affect ichthyogeographic regions, which have implication for species conservation and management. The manuscript is well written, structured and easy to read, though there are some sections that need some improvement. This is the case of the Abstract (particularly the last part – see below) and the Results: authors should not use references in this section, so all comparisons and judgements should be saved to the Discussion (describe only your main results here). Below there are some specific comments that authors could also address to improve their manuscript.

SPECIFIC COMMENTS

Line 23-24 – “the current biogeographic freshwater ecoregion scheme”. Not clear. Could you clarify? Any particular ecoregion?

Line 24-25 – “pre-PSC taxonomy period also show four ichthyogeographic division in Greece”. But you say on line 21 that only 2 divisions were found.

Line 25-29 – This last part of the Abstract is poor and needs significant improvement: i) you should not end an abstract stating an objective of the study (lines 28-29). Rather ii) the abstract should end by stating the implications of this study not only for Greece but also for other contexts worldwide. “How can the findings of my study be useful to others elsewhere?”. This is the questions you should answer at the end of your abstract, otherwise it turns too local and may have few interest to readers.

Line 52 – Comma after “organisms”.

Line 67 – Comma after “2007”.

Line 89 – Replace “to the new might”, by “to the new one, might”

Line 100-101- Perhaps this is a good sentence to place at the end of the abstract (implications elsewhere) according to my comment from lines 25-29.

Line 122 – Add “of Greece” after “units”.

Figure 1 – The figure does not have a scale neither the indication of the north. Please add them both.

Line 123 – This table (limits are lacking) could be added as Supplementary Material.

Line 126 – Comma after “analysis”.

Line 143 – Refer to the purpose of conducting a simper analysis. I suppose to find indicator species. Also include a reference.

Line 144 – Add reference for the Primer 6.

Line 146 and throughout the manuscript – Please be consistent in the use of bibliographic source. Use numbers between square brackets as the journal demands.

Line 148 – These questions that you will address on the results, should correspond to your objectives, outlined at the end of the Introduction.

Line 156 – Please refer the name of the study area (Greece) to where this applies.

Line 162 – Comma after “these”.

Line 164 and throughout the Results section: References should not be used in the Results sections. Make the comparisons and judgements on the Discussion.

Line 166- “southern Greece”?

Line 171-173 – I think you should give names to your cluster, based on these ecoregions; The same on figure 3a, i.e. northern Greece and southern Greece.

Line 184-185 – Do not understand this sentence; again, references should not be used here. Suggest remove the sentence.

Line 187 – Which ones?

Line 189 –“region of the Adriatic”. To which number corresponds on the map?

Line 184-198 – Re-write this section, by removing any references/judgements/comparisons. Describe your main results only!

Line 199 – What was the purpose of using Simper analysis? Refer in M&M.

Line 199 – “high dissimilarity values”. But on line 204 (caption to Table 2) you speak of similarity instead. It would be useful to have both similarity within Ichthyoregions and dissimilarity between Ichthyoregions.

Line 203 – Not clear “conditions” in this context.

Line 207 – please avoid these long conditional titles. Give concise, clear and short sub-titles.

Line 208 – dataset, not data-set.

Author Response

REVIEWER 3.

We thank this author for his kind words about the clarity of the paper and its importance. We were eager to improve various parts of the paper and have made major revisions, e.g. parts of the Abstract were rephrased as suggested by this reviewer.

The results section has been re-done and nearly all refs have been deleted. One slight point of divergence from the norm in results sections; we feel we must add one ref to present specific and local details and aspects of the analyses. In fact we have retained only  Zogaris and Economou 2017 as a baseline biogeographical delineation that helps orientate the reader with respect to our immediate results; and this map is re-mapped in Figure 4 in the discussion as well (along with other 3 maps). (In our case- there are published evidence of differences, gaps  etc- we have included this reference). We understand this is not commonplace in a typical results section, but in this case these minute nuances would be lost or would crowd-in the discussion.

We have clarified as much as possible based on this reviewers careful comments and suggestions including:

Line 23-24 – “the current biogeographic freshwater ecoregion scheme”. Not clear. Could you clarify? Any particular ecoregion?

-        Has been re-phrased.

Line 24-25 – “pre-PSC taxonomy period also show four ichthyogeographic division in Greece”. But you say on line 21 that only 2 divisions were found.

-        Has been re-phrased.

Line 25-29 – This last part of the Abstract is poor and needs significant improvement: i) you should not end an abstract stating an objective of the study (lines 28-29). Rather ii) the abstract should end by stating the implications of this study not only for Greece but also for other contexts worldwide. “How can the findings of my study be useful to others elsewhere?”. This is the questions you should answer at the end of your abstract, otherwise it turns too local and may have few interest to readers.

-The abstract has been re-phrased and specific conclusions added exactly as suggested by this author.

Line 52 – Comma after “organisms”.

-OK

Line 67 – Comma after “2007”.

-OK

Line 89 – Replace “to the new might”, by “to the new one, might”

-Restructured phrasing

Line 100-101- Perhaps this is a good sentence to place at the end of the abstract (implications elsewhere) according to my comment from lines 25-29.

-Done

Line 122 – Add “of Greece” after “units”.

-Done

Figure 1 – The figure does not have a scale neither the indication of the north. Please add them both.

-Scale and orientation added. 

Line 123 – This table (limits are lacking) could be added as Supplementary Material.

-We prefere the table accompany orientation map please.

Line 126 – Comma after “analysis”.

-Done

Line 143 – Refer to the purpose of conducting a simper analysis. I suppose to find indicator species. Also include a reference.

-Rephrased and details on SIMPER provided as proposed by other reviewers as well.

Line 144 – Add reference for the Primer 6.

-Done

Line 146 and throughout the manuscript – Please be consistent in the use of bibliographic source. Use numbers between square brackets as the journal demands.

-        This has been ameliorated; in a few instance we want the specific authors and classical work to stand out in the text and this is why we show the author names and/or dates – to provide historical context.

Line 148 – These questions that you will address on the results, should correspond to your objectives, outlined at the end of the Introduction.

-Roughly they do and they are placed as questions in order to guide and orientate the reader.

Line 156 – Please refer the name of the study area (Greece) to where this applies.

-Done and rephrased

Line 162 – Comma after “these”.

-Done and rephrased

Line 164 and throughout the Results section: References should not be used in the Results sections. Make the comparisons and judgements on the Discussion.

- The results section has been re-done and nearly all refs have been deleted. One slight point of divergence from the norm in results sections; we feel we must add one ref to present specific and local details and aspects of the analyses. In fact we have retained only  Zogaris and Economou 2017 as a baseline biogeographical delineation that helps orientate the reader with respect to our immediate results; and this map is re-mapped in Figure 4 in the discussion as well (along with other 3 maps). (In our case- there are published evidence of differences, gaps  etc- we have included this reference). We understand this is not commonplace in a typical results section, but in this case these minute nuances would be lost or would crowd-in the discussion.

Line 166- “southern Greece”?

-        Corrected to northern Aegean and this is further rephrased and corrected in the legend of Figure 3.

Line 171-173 – I think you should give names to your cluster, based on these ecoregions; The same on figure 3a, i.e. northern Greece and southern Greece.

-        Done! Excellent point: corrected in the legend of Figure 3.

Line 184-185 – Do not understand this sentence; again, references should not be used here. Suggest remove the sentence.

-        Sent to Discussion as suggested above.

Line 187 – Which ones?

-        Sent to Discussion as suggested above.

Line 189 –“region of the Adriatic”. To which number corresponds on the map?

-        Rephrapsed in Discussion as suggested above.

Line 184-198 – Re-write this section, by removing any references/judgements/comparisons. Describe your main results only!

-        Done

Line 199 – What was the purpose of using Simper analysis? Refer in M&M.

-        SIMPER and ANOSIM were explained and now re-phrased in the Methods sections; they provide greater clarity of the relationships among basins, and the species involved.

Line 199 – “high dissimilarity values”. But on line 204 (caption to Table 2) you speak of similarity instead. It would be useful to have both similarity within Ichthyoregions and dissimilarity between Ichthyoregions.

-        Rephrased, we present similarity results inside icthyoregions in table 2. The most important results of dissimilarities between ichthyoregions are mentioned only in the text.

Line 203 – Not clear “conditions” in this context.

-Deleted

Line 207 – please avoid these long conditional titles. Give concise, clear and short sub-titles.

-Rephrased, shortened

Line 208 – dataset, not data-set.

-This has been rephrased; we would prefer to send the dataset to all interested parties ourselves. This reflects our concern with having poorly-trained individuals analyses the data. Of course the data are in the graphics presented – being only presence/absence data so they are not immediately required for processing the paper’s interpretation.

Round 2

Reviewer 1 Report

I believe the authors thoroughly answered to all the issues raised by the referees. The manuscript has been significantly improved. The inclusion of the ANOSIM increased the interest of the research by nicely supporting the conclusions of the authors.

I have only a few minor remarks:

- L.40: replace "hydgraphic" by "hydrographic"

- L. 98, L. 114: replace "icthyogeographical" by "ichthyogeographical"

- L. 113: replace "regionization" by "regionalization"

- L. 162: delete the parenthesis at the end of the sentence

- L. 164: replace "icthyofauna" by "ichthyofauna"

- L. 191: replace "Economidis's" by "Economidis' "

- L. 204-205: "ANOSIM results between ichthyoregions are shown in table 2, only for the new
 taxonomy, since the old taxonomy cluster revealed no more than two clusters". In fact, I don't understand why your cluster 2 in Figure 3a could not be split in several subclusters (as you have done in Fig.3b)

- L. 217: replace "sub cluster" by "subcluster"

- L. 217: replace "Squalius peloponensis" by "Squalius peloponnensis"

- L. 223: replace "similariy" by "similarity"

- L. 229: Replace "in the parenthesis" by "in the parentheses"

- L. 306: "and this pattern of relatedness". I suggest to delete "and"

- L. 309: Close the parenthesis (e.g. [64].

- L. 313: delete one [ and add full stop.

- L. 343: I would replace "lacustrine fisheries traditions" by "lacustrine fishery traditions"

- L. 361: replace "into sub-regions" by "into subregions"

- L. 369: replace "well [e.g. [38]." by "well (e.g. [38])."

- L. 407: replace "icthyoregions" by "ichthyoregions"

- Ref. 4 (L. 433-436): write "accompagné" and "de latitude boréale"

-. L. 451: replace "poisons" by "poissons"

- Ref. 23: use lowercase letters for all the author names

- References: Write genus and species names in italics when necessary (e.g. ref 39, 67, 69, 87).

Author Response

I believe the authors thoroughly answered to all the issues raised by the referees. The manuscript has been significantly improved. The inclusion of the ANOSIM increased the interest of the research by nicely supporting the conclusions of the authors. 

I have only a few minor remarks:

- L.40: replace "hydgraphic" by "hydrographic"

Applied in the text

- L. 98, L. 114: replace "icthyogeographical" by "ichthyogeographical"

Applied in the text

- L. 113: replace "regionization" by "regionalization"

Applied in the text

- L. 162: delete the parenthesis at the end of the sentence

Applied in the text

- L. 164: replace "icthyofauna" by "ichthyofauna"

Whole ichthyfauna nomenclature phrase have changed with the word “taxonomy”

- L. 191: replace "Economidis's" by "Economidis' "

Corrected

- L. 204-205: "ANOSIM results between ichthyoregions are shown in table 2, only for the new
 taxonomy, since the old taxonomy cluster revealed no more than two clusters". In fact, I don't understand why your cluster 2 in Figure 3a could not be split in several subclusters (as you have done in Fig.3b)

Although our scope is not to delineate new ichthyoregions we have applied the analysis in order to provide a holistic interpretation of the classification results. Since most of the subclusters are separated mainly due to low species richness we didn’t add an extra table in the text. We just refer to the analysis in the text and explain this. We feel this treatment and explanation is enough to best showcase the results. 

- L. 217: replace "sub cluster" by "subcluster"

Corrected

- L. 217: replace "Squalius peloponensis" by "Squalius peloponnensis"  

We retain S. peloponensis as given in all modern accounts (Barbieri et al. 2015, Kottelat & Freyhof 2007).

- L. 223: replace "similariy" by "similarity"

Corrected

- L. 229: Replace "in the parenthesis" by "in the parentheses"

Corrected

- L. 306: "and this pattern of relatedness". I suggest to delete "and" 

Deleted

- L. 309: Close the parenthesis (e.g. [64]. 

Corrected

- L. 313: delete one [ and add full stop.

Corrected

- L. 343: I would replace "lacustrine fisheries traditions" by "lacustrine fishery traditions"

Replaced in the text

- L. 361: replace "into sub-regions" by "into subregions"

Replaced in the text

- L. 369: replace "well [e.g. [38]." by "well (e.g. [38])."

Corrected

- L. 407: replace "icthyoregions" by "ichthyoregions"

Replaced in the text

- Ref. 4 (L. 433-436): write "accompagné" and "de latitude boréale"

Corrected

-. L. 451: replace "poisons" by "poissons"

Replaced

- Ref. 23: use lowercase letters for all the author names

Corrected

- References: Write genus and species names in italics when necessary (e.g. ref 39, 67, 69, 87). –

Corrected

All of the above have been corrected exactly as proposed by the reviewer; we thank him/her for meticulous screening of the text and references. Finally, one of the authors is a native Canadian English speaker and he has reviewed with the care the text. His cousin, Dr. Jim Manolis (The Nature Conservancy, Minnesota) has also reviewed the text for English.   
